# Learning Latent Representations in Neural Networks for Clustering through Pseudo Supervision and Graph-based Activity Regularization

**Ozsel Kilinc**
Electrical Engineering Department
University of South Florida
Tampa, FL 33620
ozsel@mail.usf.edu

**Ismai Uysal**
Electrical Engineering Department
University of South Florida
Tampa, FL 33620
iuysal@usf.edu

## Abstract

In this paper, we propose a novel unsupervised clustering approach exploiting the hidden information that is indirectly introduced through a pseudo classification objective. Specifically, we randomly assign a pseudo parent-class label to each observation which is then modified by applying the domain specific transformation associated with the assigned label. Generated pseudo observation-label pairs are subsequently used to train a neural network with Auto-clustering Output Layer (ACOL) that introduces multiple softmax nodes for each pseudo parent-class. Due to the unsupervised objective based on Graph-based Activity Regularization (GAR) terms, softmax duplicates of each parent-class are specialized as the hidden information captured through the help of domain specific transformations is propagated during training. Ultimately we obtain a $k$-means friendly latent representation. Furthermore, we demonstrate how the chosen transformation type impacts performance and helps propagate the latent information that is useful in revealing unknown clusters. Our results show state-of-the-art performance for unsupervised clustering tasks on MNIST, SVHN and USPS datasets, with the highest accuracies reported to date in the literature.

## 1 Introduction

Clustering, the unsupervised process of grouping similar examples together, is one of the most fundamental challenges in machine learning research and has been studied extensively in different aspects such as feature selection, distance functions, grouping methods, etc. (Aggarwal & Reddy, 2014). $k$-means (MacQueen et al., 1967) and Gaussian Mixture Models (GMM) (Bishop, 2007) are two well-known conventional clustering algorithms that are applicable to a wide range of problems. Traditionally, these methods are applied to low-level features such as raw data or gradient-orientation histograms (HOG) for images. Therefore, their distance metrics are limited to local relations in the data space and inadequate to represent hidden dependencies in latent spaces. On the other hand, spectral clustering (von Luxburg, 2007) is another conventional approach producing more flexible distance metrics than $k$-means and GMM. However, these types of solutions are not scalable to large datasets as they need to compute the full graph Laplacian matrix.

In recent years, researchers have focused on the unsupervised learning of high-level features on which to apply clustering and shown that learning good representations is important for the accuracy and robustness of the clustering task. Deep Embedding Clustering (DEC) (Xie et al., 2016) was proposed to simultaneously learn feature representations and cluster assignments using deep neural networks (DNN). In this approach, first DNN parameters are initialized with a layer-wise trained deep autoencoder (Vincent et al., 2010) and then the initialized DNN is used to obtain the latent representation on which to perform $k$-means clustering for the initialization of cluster centers. This complicated initialization is followed by a challenging optimization process that minimizes the Kullback–Leibler (KL) divergence between the centroid-based probability distribution and the

auxiliary target distribution derived from the soft cluster assignments. Similarly, Joint Unsupervised Learning (JULE) (Yang et al., 2016) combines agglomerative clustering with convolutional neural networks (CNN) and formulates them as a recurrent process. Although JULE proposes an end-to-end learning framework, it suffers scalability issues due to its agglomerative clustering.

Novel deep generative models that can be trained via direct backpropagation have recently been proposed avoiding the difficulties in preexisting generative models such as Restricted Boltzmann Machines (RBM), Deep Belief Networks (DBN) and Deep Boltzmann Machines (DBM) that are trained by MCMC-based algorithms (Hinton et al., 2006; Salakhutdinov & Hinton, 2009). Among two canonical examples of these models, Variational Autoencoders (VAE) (Kingma & Welling, 2013; Rezende et al., 2014) integrate stochastic latent variables into the conventional autoencoder architecture while Generative Adversarial Networks (GAN) (Goodfellow et al., 2014) propose an adversarial training procedure implementing a min-max adversarial game between two neural networks: the discriminator and the generator. Following these advances, researchers have started to study new hybrid models with the goal of performing unsupervised clustering through deep generative models. For example, Variational Deep Embedding (VaDE) (Jiang et al., 2017) proposed a clustering framework combining VAE and GMM together. Also, Gaussian Mixture Variational Autoencoder (GMVAE) (Dilokthanakul et al., 2016) built upon the semi-supervised model by Kingma et al. (2014) to perform unsupervised clustering within the VAE framework with a Gaussian mixture as a prior distribution. GAN-based methods include: Categorical Generative Adversarial Networks (CatGAN) (Springenberg, 2015), an approach incorporating neural network classifiers with an adversarial generative model, and Adversarial Autoencoder (AAE) (Makhzani et al., 2015), a probabilistic autoencoder variant integrating traditional reconstruction error with adversarial training criterion of GANs. Besides, Premachandran & Yuille (2016) proposes to fuse the disentangled features learned by Information Maximizing Generative Adversarial Networks (InfoGAN), an extension to GANs that uses mutual information to induce representation, with $k$-means clustering.

In this paper, we propose a novel unsupervised clustering approach building upon the previous study on learning of latent annotations in a particular semi-supervised setting where a coarse level of supervision is available for all observations, i.e. parent-class labels, but the model has to learn a fine level of latent annotations, i.e. sub-classes, under each one of these parents. For clarification, assume that we are given a dataset of hand-written digits such as MNIST (LeCun et al., 1998) where the overall task is the complete categorization of each digit, but the only available supervision is whether a digit is smaller or greater than 5. To study this particular semi-supervised setting on neural networks, Kilinc & Uysal (2017a) proposed a novel output layer modification, Auto-clustering Output Layer (ACOL). ACOL allows simultaneous supervised classification (per provided parent-classes) and unsupervised clustering (within each parent) where clustering is performed through Graph-based Activity Regularization (GAR) technique recently proposed in Kilinc & Uysal (2017b). More specifically, as ACOL duplicates the softmax nodes at the output layer for each class, GAR allows for competitive learning between these duplicates on a traditional error-correction learning framework.

To learn latent annotations in a fully unsupervised setup, we substitute the real, yet unavailable, parent-class information with a pseudo one. More specifically, we choose a domain specific transformation to be applied to the observations in a dataset to generate examples for a pseudo parent-class. The transformed dataset constitutes the examples of that pseudo parent-class and every new transformation generates a new one. Regarding the MNIST example for this fully unsupervised setting, now we simply augment the dataset by applying a transformation to examples, e.g. rotating by $90^o$, and label transformed examples as *rotated* and non-transformed examples as *original*. This new augmented dataset is provided to the network as a two-class classification problem with pseudo classes labeled as *original* and *rotated* as visualized in Figure 1. While being trained over this pseudo supervision, through ACOL and GAR, the neural network learns the latent representation distinguishing the real digit identities in an unsupervised manner.

The idea of employing an auxiliary task to learn a good data representation has been previously studied for different domains (Collobert et al., 2011; Ahmed et al., 2008). Most recent study, Exemplar CNN (Dosovitskiy et al., 2016), proposed to use a regularizer enforcing the feature representation to be approximately invariant to the transformations while training the network to discriminate between a set of pseudo parent-classes ("surrogate classes" with their definition). This approach requires thousands of transformations to obtain a good representation and also it cannot exploit more than 300 examples per "surrogate class" severely limiting its scalability. Furthermore, some elementary

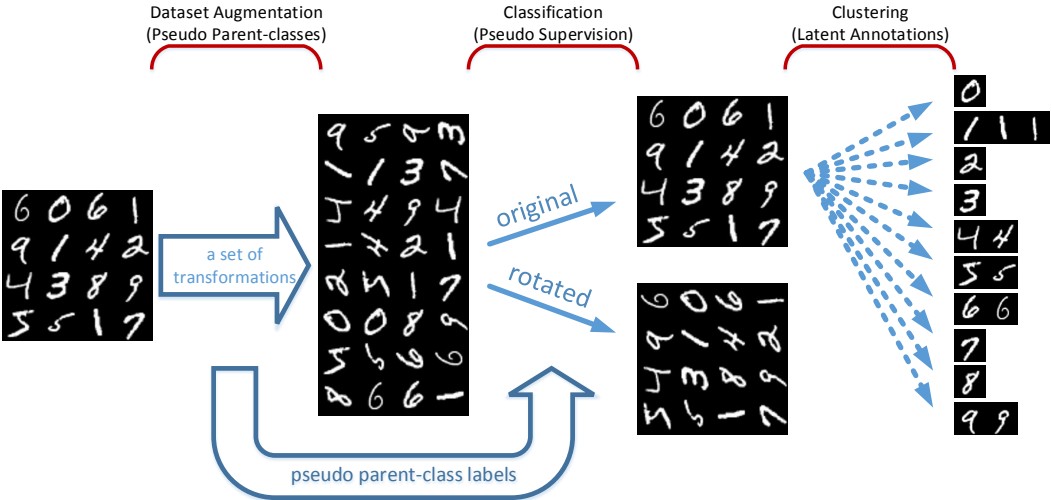

Figure 1: Assume that we are given a dataset of hand-written digits such as MNIST where the overall task is the complete categorization of each digit. Then, we simply augment the dataset by applying a transformation to examples, e.g. rotating by $90^o$, and label each of them either as *original* or as *rotated*. This new augmented dataset is provided to the network as a two-class classification problem. While being trained over this pseudo supervision, through ACOL and GAR, the neural network also learns the latent representation distinguishing the real digit identities in unsupervised an manner.

transformations, such as rotation, have only a minor impact on the performance. In comparison, in our approach, only 8 pseudo parent-classes generated by rotation-based transformations provide a rich latent representation to obtain state-of-the-art unsupervised clustering performance.

## 2 BACKGROUND

### 2.1 AUTO-CLUSTERING OUTPUT LAYER

Unlike traditional output layer structure, the Auto-clustering Output Layer (ACOL) (Kilinc & Uysal, 2017a) defines more than one softmax node ($k_s$ duplicates) per parent-class. Outputs of $k_s$ duplicated softmax nodes that belong to the same parent are then combined in a subsequent pooling layer for the final prediction. Training is performed in the configuration shown in Figure 2 where $n_p$ is the number of parent-classes. This might look like a classifier with redundant softmax nodes. However, duplicated softmax nodes of each parent are specialized using GAR throughout the training in a way that each one of $n = n_p k_s$ softmax nodes represent an individual sub-class of a parent, i.e. annotation.

In order to mathematically describe this modification, let us consider a neural network with $L-1$ hidden layers where $l$ denotes the individual index for each layer such that $l \in \{0, ..., L\}$. Let $\boldsymbol{Y}^{(l)}$ denote the output of the nodes at layer $l$. $\boldsymbol{Y}^{(0)} = \boldsymbol{X}$ is the input and $f(\boldsymbol{X}) = f^{(L)}(\boldsymbol{X}) = \boldsymbol{Y}^{(L)} = \boldsymbol{Y}$ is the output of the entire network. $\boldsymbol{W}^{(l)}$ and $\mathbf{b}^{(l)}$ are the weights and biases of layer $l$, respectively. Then, the feedforward operation of the neural networks can be written as

$$f^{(l)}(\boldsymbol{X}) = \boldsymbol{Y}^{(l)} = h^{(l)}(\boldsymbol{Y}^{(l-1)}\boldsymbol{W}^{(l)} + \boldsymbol{b}^{(l)}) \tag{1}$$

where $h^{(l)}(.)$ is the activation function applied at layer $l$.

For ACOL networks, $h^{(L-1)}(.)$ and $h^{(L)}(.)$ respectively correspond to softmax and linear activation functions. Also, $\boldsymbol{W}^{(L)} := [\boldsymbol{I}_{n_p} \ldots \boldsymbol{I}_{n_p}]^T$ and $\boldsymbol{b}^{(L)} := \boldsymbol{0}$ where $\boldsymbol{I}$ denotes the identity matrix as ACOL simply defines constant weights between the augmented softmax layer and the pooling layer to sum up the output probabilities of the softmax nodes belonging to the same parent. Let $\boldsymbol{Z}$ denote the activities at the input of augmented softmax layer such that

$$\boldsymbol{Z} := \boldsymbol{Y}^{(L-2)}\boldsymbol{W}^{(L-1)} + \boldsymbol{b}^{(L-1)} \tag{2}$$

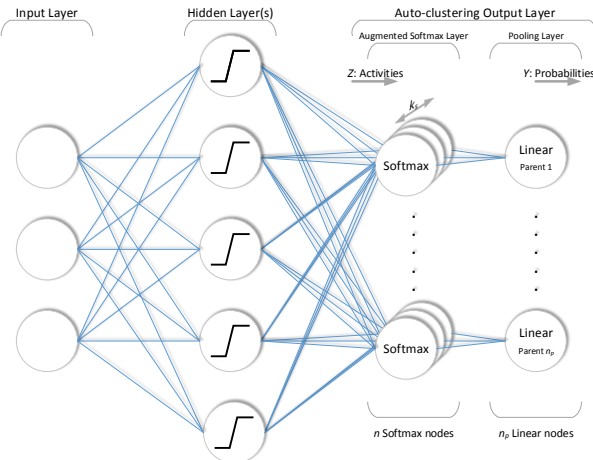

Figure 2: Neural network structure with the ACOL. Each softmax node corresponds to an individual sub-class of a parent, i.e. annotation. During feedforward operation of the network, pooling layer calculates final parent-class predictions through sub-class probabilities.

corresponding to an $m \times n$ matrix where $m$ is the number of examples and $n$ is the total number of all softmax nodes at the augmented softmax layer such that $n = n_p k_s$, where $n_p$ is the number of parent-classes and $k_s$ is the clustering coefficient of ACOL. Then, the output of the ACOL applied network can be written in terms of $\boldsymbol{Z}$ as

$$\boldsymbol{Y} = \text{softmax}\left(\boldsymbol{Z}\right)\boldsymbol{W}^{(L)} \tag{3}$$

## 2.2 Graph-based Activity Regularization

Kilinc & Uysal (2017a) adopted the Graph-based Activity Regularization (GAR) technique (Kilinc & Uysal, 2017b) as the unsupervised regularization term to create competition between the duplicated softmax nodes of the augmented softmax layer which ultimately results in specialized but equally-active softmax nodes each representing a latent annotation within a parent.

The GAR technique applies the regularization over the positive part of the activities at the input of softmax nodes such that

$$g\left(\boldsymbol{X}\right) = \boldsymbol{B} := \max\left(\boldsymbol{0}, \boldsymbol{Z}\right) \tag{4}$$

and defines two terms to turn $n \times n$ symmetric matrix $\boldsymbol{N}$, which is defined as $\boldsymbol{N} := \boldsymbol{B}^T \boldsymbol{B}$, into the identity matrix. While the *affinity* term penalizes the non-zero off-diagonal entries of $\boldsymbol{N}$, *balance* attempts to equalize diagonal entries. Let $\boldsymbol{v}$ be a $1 \times n$ vector representing the diagonal entries of $\boldsymbol{N}$ such that $\boldsymbol{v} := [N_{11} \ldots N_{nn}]$ and $\boldsymbol{V}$ be defined as $n \times n$ symmetric matrix such that $\boldsymbol{V} := \boldsymbol{v}^T \boldsymbol{v}$. Then, the *affinity* and *balance* terms can be written as

$$\text{Affinity} = \alpha\left(\boldsymbol{B}\right) := \frac{\sum\limits_{i \neq j}^{n} N_{ij}}{(n-1)\sum\limits_{i=j}^{n} N_{ij}} \tag{5} \qquad \text{Balance} = \beta\left(\boldsymbol{B}\right) := \frac{\sum\limits_{i \neq j}^{n} V_{ij}}{(n-1)\sum\limits_{i=j}^{n} V_{ij}} \tag{6}$$

which modifies the overall objective function of the training proposed in Kilinc & Uysal (2017a) as

$$\mathcal{L}\left(f\left(\boldsymbol{X}\right), \boldsymbol{t}\right) + \mathcal{U}\left(g\left(\boldsymbol{X}\right)\right) = \mathcal{L}\left(\boldsymbol{Y}, \boldsymbol{t}\right) + c_\alpha \alpha\left(\boldsymbol{B}\right) + c_\beta\left(1 - \beta\left(\boldsymbol{B}\right)\right) + c_F ||\boldsymbol{B}||_F^2 \tag{7}$$

where $\mathcal{L}(.)$ is the supervised log loss function, $\boldsymbol{t} = [t_1 \ldots t_m]^T$ is the vector of provided parent-class labels such that $t_i \in \{1, ..., n_p\}$ (recall that, in the semi-supervised setting considered in Kilinc & Uysal (2017a), there is a real partial supervision available for all examples, e.g. a digit is smaller or greater than 5), $\mathcal{U}(.)$ is the unsupervised regularization term consisting of *affinity*, *balance* and $||\boldsymbol{B}||_F$ (the Frobenius norm for $\boldsymbol{B}$) that is employed to limit the denominators of both *affinity* and *balance* terms not to diminish their effects and $c_\alpha, c_\beta, c_F$ are the weighting coefficients.

GAR has been originally proposed for the classical type of semi-supervised setting where the number of labeled observations is much smaller than the number of unlabeled observations, but all existing classes are equally represented by the available labels even at limited numbers. Kilinc & Uysal (2017b) have shown that defining the objective of the regularization over the matrix $N$ yields a scalable and efficient graph-based solution and that the entire operation corresponds to propagating the available labels across the graph $\mathcal{G}_{\mathcal{M}}$ whose edges are specified by the $m \times m$ symmetric matrix $M := BB^T$ that infers the adjacency of the examples based on the predictions of the neural network. More specifically, it has been shown that as the matrix $N$ turns into the identity matrix, $\mathcal{G}_{\mathcal{M}}$ becomes a disconnected graph including $n$ disjoint subgraphs each of which is $m/n$-regular. This indicates that the strong adjacencies in the matrix $M$ get stronger, weak ones diminish and each label is propagated to $m/n$ examples through the strong adjacencies.

On the other hand, in the particular semi-supervised setting considered by Kilinc & Uysal (2017a) (i.e. a coarse level of labeling is available for all observations but the model still needs to learn a fine level of latent annotation for each one of them), when applied to an ACOL network, GAR provides that the latent information introduced by the coarse supervision is propagated from the graph $\mathcal{G}_{\mathcal{Y}}$ (whose edges are specified by $m \times m$ symmetric matrix $YY^T$) to its spanning subgraph $\mathcal{G}_{\mathcal{M}}$ to reveal deeper latent annotations. In other words, although these two graphs are made up of the same vertices ($m$ examples) while propagating the latent information that is captured through supervised adjacency introduced by $\mathcal{G}_{\mathcal{Y}}$ across $\mathcal{G}_{\mathcal{M}}$, GAR terms eliminate some of the edges of $\mathcal{G}_{\mathcal{Y}}$ from $\mathcal{G}_{\mathcal{M}}$ in a way that $\mathcal{G}_{\mathcal{M}}$ ultimately becomes a disconnected graph of $n$ disjoint subgraphs each of which now corresponds to a latent annotation.

## 3 PROPOSED FRAMEWORK

### 3.1 OBJECTIVE FUNCTION

The unsupervised clustering approach proposed in this paper adopts the same framework introduced in Kilinc & Uysal (2017a). Since the real parent-class labels (a digit is smaller or greater than 5) are unavailable in a fully unsupervised setting, we randomly assign pseudo parent-class labels each of which is associated with a domain specific transformation used to generate the examples of that pseudo parent-class.

In this setting, $n_p$ now corresponds to the number of pseudo parent-classes and $\tilde{t} = [\tilde{t}_1 \ldots \tilde{t}_m]^T$ is a vector of randomly assigned pseudo parent-class labels which are uniformly distributed across $n_p$ pseudo parent-classes such that $\tilde{t}_i \in \{1, ..., n_p\}$. Also, there exists a set of transformations $\mathcal{S}_{\mathcal{T}} = \{\mathcal{T}_1, ..., \mathcal{T}_{n_p}\}$ where transformation $\mathcal{T}_j$ is used to generate the examples of the $j^{\text{th}}$ pseudo parent-class such that $\tilde{x}_i = \mathcal{T}_j(x_i)$. $\mathcal{S}_{\mathcal{T}}$ also includes non-transformation $\mathcal{T}_1$ providing $\tilde{x}_i = \mathcal{T}_1(x_i) = x_i$ to ensure that the original observations are introduced to the network during training. $\tilde{t}$ is associated with a vector of transformations $T = [T_1 \ldots T_m]^T$ such that $T_i = \mathcal{T}_{\tilde{t}_i}$.

Let $\odot$ be an element-wise operation defined between the vector of transformations $T$ and the original input $X = [x_1 \ldots x_m]^T$ such that

$$\tilde{X} = \begin{bmatrix} \tilde{x}_1 \\ \tilde{x}_2 \\ \vdots \\ \tilde{x}_m \end{bmatrix} = T \odot X = \begin{bmatrix} T_1 \\ T_2 \\ \vdots \\ T_m \end{bmatrix} \odot \begin{bmatrix} x_1 \\ x_2 \\ \vdots \\ x_m \end{bmatrix} = \begin{bmatrix} T_1(x_1) \\ T_2(x_2) \\ \vdots \\ T_m(x_m) \end{bmatrix} = \begin{bmatrix} \mathcal{T}_{\tilde{t}_1}(x_1) \\ \mathcal{T}_{\tilde{t}_2}(x_2) \\ \vdots \\ \mathcal{T}_{\tilde{t}_m}(x_m) \end{bmatrix} \tag{8}$$

where $\tilde{X}$ corresponds to the modified input per randomly assigned pseudo labels $\tilde{t}$. The output of the entire network and the positive part of the augmented softmax layer activities respectively become $Y = f(\tilde{X})$ and $B = g(\tilde{X})$. Then, the objective function defined in (7) can simply be adopted by substituting the real, yet unavailable, observation-label pair $(X, t)$ with a pseudo one $(\tilde{X}, \tilde{t})$ such that

$$\mathcal{L}\big(f(\tilde{X}), \tilde{t}\big) + \mathcal{U}\big(g(\tilde{X})\big) = \mathcal{L}(Y, \tilde{t}) + c_\alpha \alpha(B) + c_\beta \big(1 - \beta(B)\big) + c_F ||B||_F^2 \tag{9}$$

## 3.2 Modified *Affinity* and *Balance* Terms

Recall that an $n \times n$ symmetric matrix $\boldsymbol{N} = \boldsymbol{B}^T \boldsymbol{B}$ specifies the edges of the graph between the softmax duplicates and that GAR terms have been proposed to regularize the matrix $\boldsymbol{N}$ in a way that it turns into the identity matrix. While the objective of *affinity*, i.e. penalizing the non-zero off-diagonal entries of $\boldsymbol{N}$, corresponds to assigning an example to only one softmax node with the probability of 1, the objective of *balance*, i.e. equalizing diagonal entries of $\boldsymbol{N}$, corresponds to preventing collapsing onto a subspace of dimension less than $n$.

Among the off-diagonal entries of $\boldsymbol{N}$ determining the *affinity* cost, for each one of $n$ softmax nodes, there exist $k_s - 1$ entries describing its relation with the other duplicates of the same parent-class (let us define them as *intra-parent* entries) and $(n_p - 1)k_s$ entries describing its relation with the softmax nodes belonging to other parent-classes (let us define them as *inter-parent* entries). While *inter-parent* entries are explicitly affected by the pseudo classification objective as well as the regularization, *intra-parent* entries do not experience the classification directly. Therefore, the *affinity* cost due to *inter-parent* entries is minimized at a different rate than the *affinity* cost due to *intra-parent* entries. On the other hand, as it is calculated over the diagonal entries of $\boldsymbol{N}$, the *balance* cost does not either experience the pseudo classification objective explicitly. As a result, due to the direct impact of the pseudo classification objective which is observed only on the *affinity* cost, the weighting between the regularization terms actively alters during the training and needs to be re-tuned through the hyperparameters $c_\alpha$ and $c_\beta$. This effect can be observed more clearly as $n_p$, the number of parent-classes, increases.

To ensure a more robust regularization we introduce a modification for the *affinity* and *balance* terms: We discard all *inter-parent* entries of $\boldsymbol{N}$ and represent the remaining ones as a three dimensional tensor $\tilde{\boldsymbol{N}}$. Thus, $\tilde{\boldsymbol{N}}$ is a $k_s \times k_s \times n_p$ tensor such that $\tilde{\boldsymbol{N}}_{:,:,k}$ specifies the relations between $k_s$ softmax duplicates of the $k^{\text{th}}$ parent-class where $k \in \{1, ..., n_p\}$. Also, $\tilde{\boldsymbol{V}}$ is another $k_s \times k_s \times n_p$ tensor defined as

$$\tilde{\boldsymbol{V}}_{:,:,k} = [\tilde{N}_{1,1,k} \ldots \tilde{N}_{k_s,k_s,k}]^T [\tilde{N}_{1,1,k} \ldots \tilde{N}_{k_s,k_s,k}] \tag{10}$$

Then, the modified *affinity* and *balance* terms can be respectively written as

$$\tilde{\alpha}(\boldsymbol{B}) := \frac{1}{n_p} \sum_{k=1}^{n_p} \frac{\sum_{i \neq j}^{k_s} \tilde{N}_{ijk}}{(k_s - 1) \sum_{i=j}^{k_s} \tilde{N}_{ijk}} \tag{11} \qquad \tilde{\beta}(\boldsymbol{B}) := \frac{1}{n_p} \sum_{k=1}^{n_p} \frac{\sum_{i \neq j}^{k_s} \tilde{V}_{ijk}}{(k_s - 1) \sum_{i=j}^{k_s} \tilde{V}_{ijk}} \tag{12}$$

and simply correspond to calculating the original terms given in (5), (6) on each 2-D $k_s \times k_s \times 1$ slice of $\tilde{\boldsymbol{N}}$ and $\tilde{\boldsymbol{V}}$ tensors and then averaging the results for $n_p$ of them.

Replacing these modified terms in (9), the overall modified objective function becomes

$$\mathcal{L}\big(f(\tilde{\boldsymbol{X}}), \tilde{\boldsymbol{t}}\big) + \mathcal{U}\big(g(\tilde{\boldsymbol{X}})\big) = \mathcal{L}\big(\boldsymbol{Y}, \tilde{\boldsymbol{t}}\big) + c_\alpha \tilde{\alpha}(\boldsymbol{B}) + c_\beta \big(1 - \tilde{\beta}(\boldsymbol{B})\big) + c_F ||\boldsymbol{B}||_F^2 \tag{13}$$

## 3.3 Training and Cluster Assignments

Network parameters are trained by implementing the stochastic optimization method Adam (Kingma & Ba, 2014) based on the objective given in (13). After training, $k$-means clustering is performed on the representation space observed in the hidden layer preceding the augmented softmax layer such that

$$\boldsymbol{F} = \boldsymbol{Y}^{(L-2)} = f^{(L-2)}(\boldsymbol{X}) \tag{14}$$

Recalling that the original examples are already introduced to the network as the examples of first pseudo parent-class through transformation $\mathcal{T}_1$, we obtain the latent space representation only for the original examples to perform $k$-means clustering.

One might suggest performing $k$-means clustering on the representation observed in the augmented softmax layer ($\boldsymbol{Z}$ or $\text{softmax}(\boldsymbol{Z})$) rather than $\boldsymbol{F}$. Properties and respective clustering performances of these representation spaces are empirically demonstrated in the following sections.

Algorithm 1 below describes the entire training and cluster assignment procedure.

---

**Algorithm 1:** Model training and cluster assignments

---

**Input** : $X = [x_1 \dots x_m]^T, n_p,$
      a set of transformations $\mathcal{S}_{\mathcal{T}} = \{\mathcal{T}_1, ..., \mathcal{T}_{n_p}\}$,
      batch size $b$, weighing coefficients $c_\alpha, c_\beta, c_F$, the number of clusters $k$

**repeat**

    $\tilde{t} \longleftarrow \text{random}(n_p)$                // Randomly assign labels across $n_p$ classes
    $T \longleftarrow [\mathcal{T}_{\tilde{t}_1}, ..., \mathcal{T}_{\tilde{t}_m}]$     // Obtain the vector of transformations corresponding
     to $\tilde{t}$
    $\tilde{X} \longleftarrow T \odot X$                         // Obtain the modified input
    $\{(\acute{X}_1, \acute{t}_1), ..., (\acute{X}_{m/b}, \acute{t}_{m/b})\} \longleftarrow (\tilde{X}, \tilde{t})$     // Shuffle and create batch pairs
    **for** $i \leftarrow 1$ **to** $^m/_b$ **do**
        Take $i^{\text{th}}$ pair $(\acute{X}_i, \acute{t}_i)$
        Forward propagate for $\acute{Y}_i = f(\acute{X}_i)$ and $\acute{B}_i = g(\acute{X}_i)$
        Take a gradient step for $\mathcal{L}(\acute{Y}_i, \acute{t}_i) + c_\alpha \tilde{\alpha}(\acute{B}_i) + c_\beta (1 - \tilde{\beta}(\acute{B}_i)) + c_F ||\acute{B}_i||_F^2$

**until** stopping criteria is met

$F \longleftarrow f^{(L-2)}(X)$     // Obtain latent space representation $F$ for the original examples
$y \longleftarrow \text{kmeans}(F, k)$           // Assign clusters by performing $k$-means on $F$

**return** : Cluster assignments $y$

---

# 4 EXPERIMENTS

## 4.1 EXPERIMENTAL SETUP AND DATASETS

The models have been implemented in Python using Keras (Chollet, 2015) and Theano (Theano Development Team, 2016). Open source code is available at http://github.com/ozcell/LALNets that can be used to reproduce the experimental results obtained on three benchmark image datasets, MNIST (LeCun et al., 1998), SVHN (Netzer et al., 2011) and USPS. Specifications of these datasets are presented in Table 1.

Table 1: Datasets used in the experiments.

| | Data type | Number of examples | Dimension | Number of classes |
|---|---|---|---|---|
| MNIST | Image: Hand-written digits | Train: 60000, Test: 10000 | $1 \times 28 \times 28$ | 10 |
| USPS | Image: Hand-written digits | Train: 7291, Test: 2007 | $1 \times 16 \times 16$ | 10 |
| SVHN | Image: Street-view digits | Train: 73257, Test: 26032 | $3 \times 32 \times 32$ | 10 |

All experiments have been performed on a 6-layer convolutional neural network (CNN) model whose specifications are given in Table 2 where coefficients of GAR terms have been chosen as $k_s = 20, c_\alpha = 0.1, c_\beta = 1, c_F = 0.000001$. During training, pseudo supervised objective is introduced as an 8 pseudo parent-class classification problem, i.e. $n_p = 8$, through the following rotation-based transformations:

$$\mathcal{T}_i = \begin{cases} i = 1: & \text{No transformation} \\ i = 2: & \text{Rotate by } 90^o \\ i = 3: & \text{Rotate by } 180^o \\ i = 4: & \text{Rotate by } 270^o \\ i = 5: & \text{Flip horizontally} \\ i = 6: & \text{Flip horizontally + Rotate by } 90^o \\ i = 7: & \text{Flip horizontally + Rotate by } 180^o \\ i = 8: & \text{Flip horizontally + Rotate by } 270^o \end{cases} \quad (15)$$

For all experiments, we used a batch size of 400 and each experiment has been repeated 10 times. To ensure that the representation obtained through the proposed approach is well-generalized for never-seen-before data, we train the neural network parameters using only the training set examples

of each dataset and obtain the clustering performances using $k$-means with $k = 10$ on the latent space representation $F$ of the untransformed test set examples (through $\mathcal{T}_1$).

Table 2: Specifications of the CNN model used in the experiments.

| Model name | Specification |
|---|---|
| 6-layer CNN | 2*(32x3x3) - MP2x2 - Drop(0.2) - 2*(64x3x3) - MP2x2 - Drop(0.3) - FC 2048 - Drop(0.5) - FC 8*20 |

## 4.2 QUANTITATIVE COMPARISON

Following Jiang et al. (2017) and Yang et al. (2016), we evaluate the test performances using unsupervised clustering accuracy given as

$$ACC = \max_{\mathfrak{f} \in \mathfrak{F}} \frac{\sum_{i=1}^{m} 1\{t_i^* = \mathfrak{f}(y_i)\}}{m} \tag{16}$$

where $t_i^*$ is the ground-truth label, $y_i$ is the assigned cluster, and $\mathfrak{F}$ is the set of all possible one-to-one mappings between assignments and labels. Both metrics range between $[0, 1]$ where a larger value indicates more precise clustering results.

Figure 3 presents the t-SNE (Maaten & Hinton, 2008) visualizations of the latent space $F$ throughout the training for 2000 untransformed test examples from MNIST. Each group corresponds to a cluster (i.e. a digit) under the first pseudo parent-class (i.e. the class of untransformed examples including all ten digits). Color codes denote the ground-truths for the digits. From epoch 1 to epoch 400 of the unsupervised (but pseudo supervised) training, clusters become well-separated and simultaneously the clustering accuracy increases. As clearly observed from this figure, using the pseudo supervision, the neural network also reveals some hidden patterns useful to distinguish the real digit identities and ultimately learns to categorize each one of them. It is also worth noting that a high level of clustering accuracy is achieved relatively quickly (after only 50 epochs) as seen both in the t-SNE and test accuracy plots.

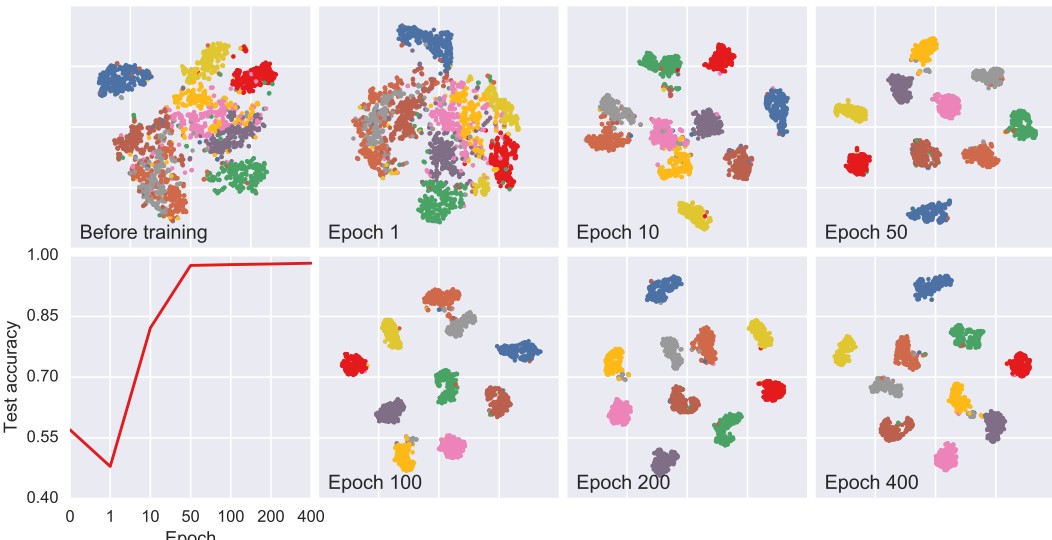

Figure 3: t-SNE visualization of the latent space $F$ throughout the training for 2000 untransformed test examples from MNIST. Color codes denote the ground-truths for the digits. Note the separation of clusters from epoch 1 to epoch 400 of the unsupervised (but pseudo supervised) training. For reference, clustering accuracy for the entire test set is also provided. This figure is best viewed in color.

Table 3 summarizes quantitative unsupervised clustering performances observed on three datasets in terms of unsupervised clustering accuracy (ACC). Results of a broad range of recent existing solu-

tions are also presented for comparison. These solutions are grouped according to their approaches to unsupervised clustering. Following the very recent developments in deep generative models, VaDE (Jiang et al., 2017) and GMVAE (Dilokthanakul et al., 2016) employ variational autoencoders while CatGAN (Springenberg, 2015), AAE (Makhzani et al., 2015) and IMSAT (Hu et al., 2017) adopt adversarial training. DEC (Xie et al., 2016) simultaneously learns feature representations and cluster assignments using DNNs. On the other hand, JULE (Yang et al., 2016) combines agglomerative clustering with CNNs. Also, the performances of two conventional approaches, applying $k$-means on raw data space and applying $k$-means on the autoencoder representation, are provided to show a baseline for unsupervised clustering performances. Our approach statistically significantly outperforms all the contemporary methods that reported unsupervised clustering performance on MNIST except IMSAT (Hu et al., 2017) displaying very competitive performance with our approach, i.e. $98.32\%(\pm0.08)$ vs. $98.40\%(\pm0.40)$. However, results obtained on the SVHN dataset, i.e. $76.80\%(\pm1.30)$ vs. $57.30\%(\pm3.90)$, show that our approach statistically significantly outperforms IMSAT on this realistic dataset and defines the current state-of-the-art for unsupervised clustering on SVHN. Besides, the USPS dataset provides another basis of comparison between our approach and JULE.

Table 3: Quantitative unsupervised clustering performance (ACC) on MNIST, USPS and SVHN datasets. Results of a broad range of recent existing solutions are also presented for comparison. The last row demonstrates the benchmark scores of the proposed framework in this article.

| | $k$ | MNIST-*test* | USPS-*full*[†] | SVHN-*test* |
|---|---|---|---|---|
| VaDE (Jiang et al., 2017) | 10 | 94.06% | - | - |
| GMVAE (Dilokthanakul et al., 2016) | 10 | 82.31%($\pm$3.75) | - | - |
| GMVAE (Dilokthanakul et al., 2016) | 16 | 87.82%($\pm$5.33) | - | - |
| GMVAE (Dilokthanakul et al., 2016) | 30 | 92.77%($\pm$1.60) | - | - |
| CatGAN (Springenberg, 2015) | 20 | 90.30% | - | - |
| AAE (Makhzani et al., 2015) | 16 | 90.45%($\pm$2.05) | - | - |
| AAE (Makhzani et al., 2015) | 30 | 95.90%($\pm$1.13) | - | - |
| IMSAT (Hu et al., 2017) | 10 | 98.40%($\pm$0.40) | - | 57.30%($\pm$3.90) |
| $k$-means (Xie et al., 2016) | 10 | 53.49% | - | - |
| AE+$k$-means (Xie et al., 2016) | 10 | 81.84% | - | - |
| DEC (Xie et al., 2016) | 10 | 84.30% | - | 11.9%($\pm$0.40)[††] |
| JULE (Yang et al., 2016) | 10 | 96.10% | 95.00% | - |
| **Our approach** | 10 | 98.32%($\pm$0.08) | 96.51%($\pm$0.26) | 76.80%($\pm$1.30) |

[†] Only for USPS dataset, following JULE (Yang et al., 2016), we reported unsupervised clustering performance over the full dataset for a fair comparison.
[††] Excerpted from (Hu et al., 2017).

## 4.3 Representation Properties

Recall that, for the 6-layer CNN model employed in the experiments, $\boldsymbol{F} = \boldsymbol{Y}^{(L-2)}$ corresponds to the output of the fully-connected layer of 2048 ReLU nodes, $\boldsymbol{Z} = \boldsymbol{F}\boldsymbol{W}^{L-1} + \boldsymbol{b}^{L-1}$ is the input of the augmented softmax layer of 160 nodes, i.e. $n = n_p k_s$, where 8 pseudo parent-classes are represented by 20 softmax duplicates each.

Figure 4 provides the average value for each dimension of $\boldsymbol{F}$, $\boldsymbol{Z}$ and softmax($\boldsymbol{Z}$) observed with respect to untransformed test set examples and the norm of the associated weights. Note that the representation on $\boldsymbol{F}$ is not distributed to the entire space but the weights associated to these unused dimensions do not decay. On the other hand, due to the pseudo supervision task, the output of the augmented softmax layer i.e. softmax($\boldsymbol{Z}$), becomes a one-hot encoded representation of which 140 dimensions, i.e. $(n_p - 1)k_s$, are inactive for the untransformed examples; however, the representation at its input is distributed to all dimensions. Figure 4 also summarizes how the dimension size of

$F$, i.e. the number of ReLU nodes in the fully-connected layer, affects the clustering performance. Decreasing the number of dimensions of $F$ up to a point, i.e. $\approx 1024$, does not significantly affect the clustering accuracy. However, further decrease beyond this point dramatically reduces the performance.

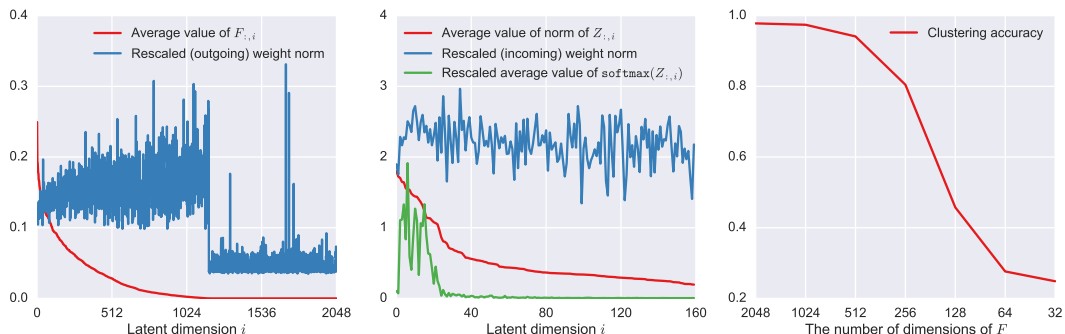

Figure 4: The average value for each dimension of $F$, $Z$ and $\mathrm{softmax}(Z)$ observed with respect to untransformed test set examples and the norm of the associated weights. Note that the representation on $F$ is not distributed to the entire space but the weights associated to these unused dimensions do not decay. On the other hand, due to the pseudo supervision task, the output of the augmented softmax layer i.e. $\mathrm{softmax}(Z)$, becomes a one-hot encoded representation of which 140 dimensions are inactive for the untransformed examples; however, the representation at its input is distributed to all dimensions. The last plot shows how the dimension size of $F$ affects the clustering performance. This figure is best viewed in color.

For comparison, Figure 5 presents t-SNE visualizations of these latent representations observed with respect to 2000 untransformed test examples from MNIST. One can clearly see that clusters are not well-separated on one-hot encoded $\mathrm{softmax}(Z)$; however, separations of the clusters are quite similar and clear on the representation spaces $F$ and $Z$. Hence, one can also obtain similar clustering accuracy, i.e. $= 98.16\% \pm (0.14)$, by applying $k$-means on the representation space $Z$.

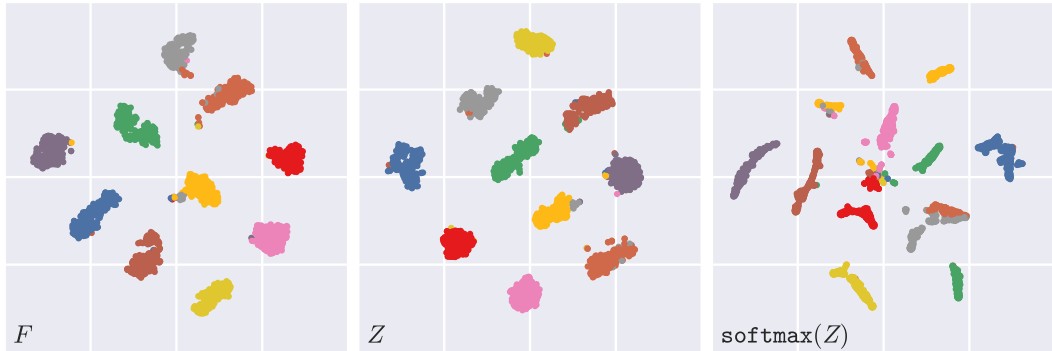

Figure 5: Comparison of t-SNE visualizations of the latent spaces $F$, $Z$ and $\mathrm{softmax}(Z)$ for 2000 test examples from MNIST. Color codes denote the ground-truths for the digits and each label represents the major digit of a cluster. Clusters are not well-separated on one-hot encoded $\mathrm{softmax}(Z)$; however, separations of the clusters are quite similar and clear on the representation spaces $F$ and $Z$. This figure is best viewed in color.

## 4.4 GRAPH INTERPRETATION OF THE LATENT INFORMATION PROPAGATION THROUGH GAR

Recall that GAR terms have been originally proposed to propagate the available labels towards the unlabeled examples in a semi-supervised setting and Kilinc & Uysal (2017a) have shown that these terms can also be adopted to propagate the hidden information that is introduced by a coarse

level of supervision and which is useful to discover a deeper level of latent annotations. In the fully unsupervised setting considered in this paper, as no real supervision is available, hidden information useful to discover unknown clusters is now captured through the help of domain specific transformations and propagated by GAR terms as well.

Figure 6 visualizes the realization of this propagation using the real predictions obtained on MNIST. Colored circles denote the ground-truths for the vertices, i.e. examples, and gray lines denote the edges, i.e. non-zero weighted connections between the examples representing their similarity. Note that, for vertices in graph $\mathcal{G}_{\mathcal{Y}}$, there are two different colors indicating true pseudo parent-class labels assigned per the applied transformation (for simplicity, out of 8, only the examples of the first two pseudo parent-classes are used for this illustration), albeit ten different colors indicating the real digit identity for vertices in graph $\mathcal{G}_{\mathcal{M}}$. Recall that edges of these two graphs, $\mathcal{E}_{\mathcal{Y}}$ and $\mathcal{E}$, are respectively inferred by matrices $\boldsymbol{Y}\boldsymbol{Y}^T$ and $\boldsymbol{B}\boldsymbol{B}^T$ where $\boldsymbol{B} = \max(0, \boldsymbol{Z})$ and that $\mathcal{G}_{\mathcal{M}}$ is the spanning subgraph of $\mathcal{G}_{\mathcal{Y}}$. That is, $\mathcal{G}_{\mathcal{M}} = (\mathcal{M}, \mathcal{E})$ shares the same vertices $\mathcal{M}$ with graph $\mathcal{G}_{\mathcal{Y}} = (\mathcal{M}, \mathcal{E}_{\mathcal{Y}})$, which is constructed per the pseudo supervision; however, $\mathcal{E}$ is a subset of $\mathcal{E}_{\mathcal{Y}}$ as some of the edges in graph $\mathcal{G}_{\mathcal{Y}}$, such as those between the examples of digit 0 and 1, are eliminated in graph $\mathcal{G}_{\mathcal{M}}$ due to GAR regularization terms. As training continues, pseudo supervision eliminates the edges between the examples of different pseudo parent-classes and turns graph $\mathcal{G}_{\mathcal{Y}}$ into a disconnected graph of $n_p = 8$ disjoint subgraphs (only two of them are illustrated). Simultaneously, GAR terms eliminate the edges between the examples of the same parent-class in graph $\mathcal{G}_{\mathcal{M}}$ to discover previously unknown clusters. Ultimately, $\mathcal{G}_{\mathcal{M}}$ becomes disconnected graphs of $\delta$ disjoint subgraphs where $n_p \leq \delta \leq n_p k_s$ and each disjoint subgraph corresponds to a cluster.

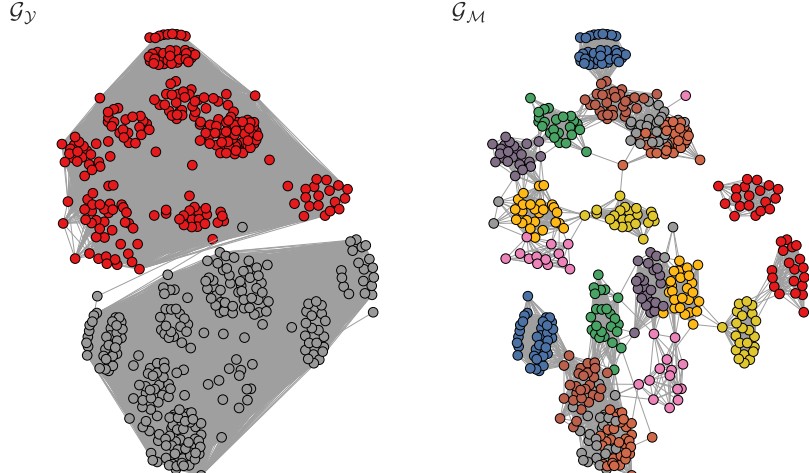

Figure 6: Visualizations of the graph $\mathcal{G}_{\mathcal{Y}}$ and its spanning subgraph $\mathcal{G}_{\mathcal{M}}$ for randomly chosen 500 test examples from MNIST (this selection is performed only for the simplicity of the visualization). Colored circles denote the ground-truths for the vertices, i.e. examples, and gray lines denote the edges, i.e. non-zero weighted connections between the examples representing their similarity. Note that, for vertices in graph $\mathcal{G}_{\mathcal{Y}}$, there are two different colors indicating true pseudo parent-class labels assigned according to the applied transformation (for simplicity, out of 8, only the examples of first two pseudo parent-classes are used for this illustration), albeit ten different colors indicating the real digit identity for vertices in graph $\mathcal{G}_{\mathcal{M}}$. As training continues, pseudo supervision eliminates the edges between the examples of different pseudo parent-classes and turns graph $\mathcal{G}_{\mathcal{Y}}$ into a disconnected graph of $n_p = 8$ disjoint subgraphs (only two of them are illustrated). Simultaneously, GAR terms eliminate the edges between the examples of the same parent-class in graph $\mathcal{G}_{\mathcal{M}}$ to discover previously unknown clusters. Ultimately, $\mathcal{G}_{\mathcal{M}}$ becomes disconnected graphs of $\delta$ disjoint subgraphs where $n_p \leq \delta \leq n_p k_s$ and each disjoint subgraph corresponds to a cluster. This figure is best viewed in color.

## 4.5 THE IMPACT OF THE NUMBER OF CLUSTERS $k$

For the quantitative clustering results, we set the number of clusters for the $k$-means to the number of classes assuming a prior knowledge, i.e. $k = 10$. To demonstrate the representation power of

the proposed approach as an unsupervised clustering model, on MNIST, we deliberately choose different $k$ values for the $k$-means clustering applied on the representation space $\boldsymbol{F}$. For two different $k$ settings i.e. 7 and 20, Figure 7 illustrates a few examples of each cluster. One can see that when $k$ is smaller than the actual number of classes, digits with similar appearances are grouped together, such as digits 4 and 9, 5 and 8, 0 and 6. When $k$ is set to a bigger value than the number of classes, some digits are divided into subclasses based on visually identifiable image properties such as digit tilt, roundness, etc. Note the differences between upright and oblique digit 1 as shown in clusters 2 and 20, between two styles of digit 6 as shown in clusters 18 and 19, and between two styles of digit 2 as shown in clusters 7 and 12.

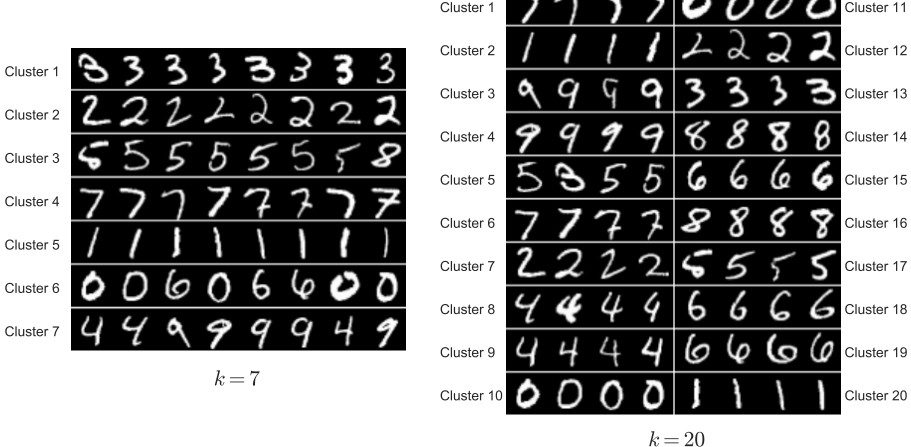

Figure 7: Illustration of a few examples of each cluster for two different $k$ settings i.e. 7 and 20. When $k$ is smaller than the actual number of classes, digits with similar appearances are grouped together, such as digits 4 and 9, 5 and 8, 0 and 6. When $k$ is set to a bigger value than the number of classes, some digits are divided into subclasses based on visually identifiable image properties such as digit tilt, roundness, etc. Note the differences between upright and oblique digit 1 as shown in clusters 2 and 20, between two styles of digit 6 as shown in clusters 18 and 19, and between two styles of digit 2 as shown in clusters 7 and 12.

## 4.6 THE IMPACT OF TRANSFORMATIONS

As the revealed unknown clusters are directly related with the captured latent information through pseudo parent-classes, choosing the right set of transformations for the clustering task of concern is crucial for the performance. Figure 8 presents t-SNE visualizations of the representation spaces observed when different sets of transformations are adopted.

The first row of Figure 8 illustrates the clustering results when one of four different transformation types, i.e. scaling, shearing, translation and random permutation of the pixels, is applied variably to generate 8 pseudo parent-classes. One can observe some level of grouping with scaling and shearing-based transformations; however, the clusters defined by these groupings do not represent real digit identities (as shown by the colored dots) and may indicate other features of images. On the other hand, translating the images or randomly permuting the pixel positions do not provide any useful knowledge to discover any well-defined clustering.

The second row of Figure 8 presents the results obtained when rotation-based transformations listed in (15) are adopted. One can easily observe that only two or four pseudo parent-classes generated using rotation-based transformations are sufficient to obtain decent clustering representing the real digit identities. Considering that, for MNIST, the clustering accuracy obtained using all 8 transformations in (15) is $98.32\%(\pm 0.08)$, we have achieved $97.80\%(\pm 0.18)$ accuracy using $\mathcal{S}_{\mathcal{T}} = \{\mathcal{T}_1, \mathcal{T}_2, \mathcal{T}_3, \mathcal{T}_4\}$, $72.52\%(\pm 6.20)$ accuracy using $\mathcal{S}_{\mathcal{T}} = \{\mathcal{T}_1, \mathcal{T}_2\}$ and $96.84\%(\pm 0.29)$ accuracy using $\mathcal{S}_{\mathcal{T}} = \{\mathcal{T}_1, \mathcal{T}_3\}$. Recalling that $\mathcal{T}_2$ and $\mathcal{T}_3$ respectively correspond to rotating the images by 90º and 180º, one can say that comparing the untransformed images with their 180º rotated versions is more effective in

terms of capturing the latent information that is useful to distinguish the real digit identities. In fact, $\mathcal{T}_3$ alone is sufficient to achieve state-of-the-art clustering accuracy on MNIST. Adding more rotation-based transformations to $\mathcal{S}_{\mathcal{T}}$ further improves the clustering performance. To summarize, the type of the transformation generating the pseudo parent-classes is more important than their number and different transformations can reveal different clustering patterns. Therefore, finding the right transformation type for the clustering task of concern is crucial for the proposed approach in this paper and it remains an important research question how to identify the kind of transformation most optimized for the clustering task at hand.

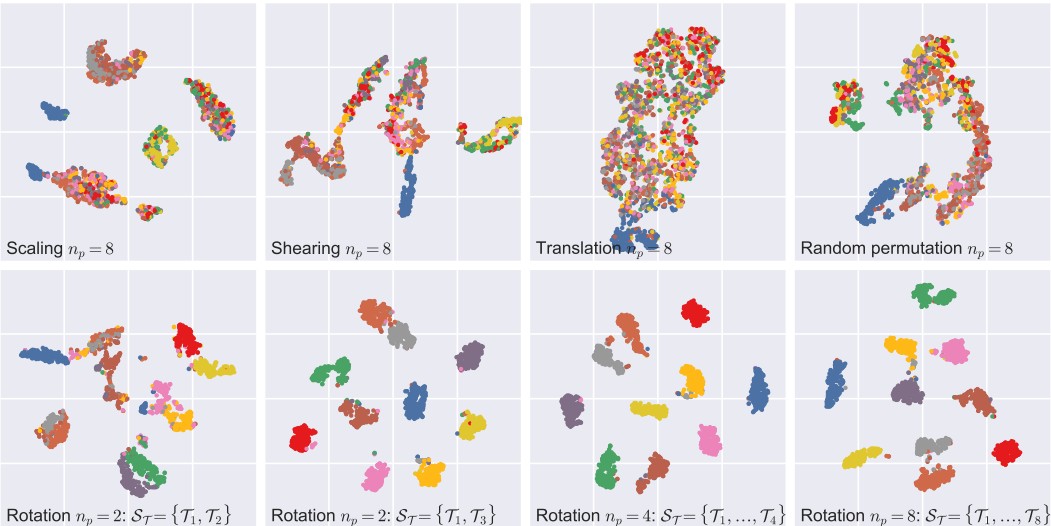

Figure 8: t-SNE visualizations of the representation spaces observed when different sets of transformations are adopted. The first row illustrates the clustering results when one of four different transformation types, i.e. scaling, shearing, translation and random permutation of the pixels, is applied variably to generate 8 pseudo parent-classes. The second row presents the results obtained when rotation-based transformations listed in (15) are adopted. To summarize, the type of the transformation generating the pseudo parent-classes is more important than their number and different transformations can reveal different clustering patterns. Therefore, finding the right transformation type for the clustering task of concern is crucial for the proposed approach in this paper.

## 5    CONCLUSION

In this paper, we introduced a novel unsupervised clustering approach building upon the previous study on an output layer modification, ACOL, which is proposed to learn latent annotations on neural networks when a partial supervision is provided. To discover unknown clusters in a fully unsupervised setup, we substitute the real, yet unavailable, partial supervision with a pseudo one. More specifically, we randomly assign pseudo parent-class labels each of which is associated with a different domain specific transformation. Each observation is modified by applying the transformation corresponding to the assigned pseudo label. Generated observation-label pairs are used to train an ACOL network that introduces multiple softmax nodes for each pseudo parent-class. Due to the unsupervised regularization based on GAR terms, each softmax duplicate under a parent-class is specialized as the latent information captured by the help of domain specific transformations is propagated throughout the training. Ultimately we obtain a $k$-means friendly latent representation. Furthermore, we demonstrate that the neural network can learn by comparing differently transformed examples and translate that knowledge to reveal unknown clusters. The proposed approach was validated on three image benchmark datasets, MNIST, SVHN and USPS, through t-SNE visualizations and unsupervised clustering accuracy exceeds those reported by well-accepted approaches in the literature. Future work will extend this approach to other domains such as sequential data. We will also explore how to optimize domain specific transformations based on known or otherwise identifiable characteristics of the dataset being considered for clustering.

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
