# OpenReview forum: "Learning Latent Representations in Neural Networks for Clustering through Pseudo Supervision and Graph-based Activity Regularization"
_ICLR.cc/2018/Conference — Accept (Poster)_

### Official Review · AnonReviewer1 · 2017-11-29
**Well written manuscript with a marginal contribution**

**Rating:** 7
**Confidence:** 3

**Review:**

The paper is well written and clear. The main idea is to exploit a schema of semisupervised learning based on ACOL and GAR for an unsupervised learning task. The idea is to introduce the notion of pseudo labelling.
Pseudo labelling can be obtained by transformations of original input data.
The key point is the definition of the transformations.
Only whether the design of transformation captures the latent representation of the input data, the pseudo-labelling might improve the performance of the unsupervised learning task.
Since it is not known in advance what might be a good set of transformations, it is not clear what is the behaviour of the model when the large portion of transformations are not encoding the latent representation of clusters.

---

> ### Author Response · Authors · 2017-12-23
> **Author feedback**
>
> We’d like to thank the reviewer for their encouraging remarks and feedback.  The reviewer makes an excellent point on the impact of transformations in clustering accuracy – specifically for datasets where domain expertise is not readily available.  We believe that for image clustering problems, the focus of this article, the proposed domain transformations work sufficiently well (state-of-the-art) based on comparative results with the recent literature on these datasets as laid out in the article.  For other domains, the exploration of the effects and optimality of transformations represents the most immediate and honestly, exciting future work which will be addressed in subsequent articles as discussed in the final section.

---

### Official Review · AnonReviewer3 · 2017-11-29
**Good paper**

**Rating:** 7
**Confidence:** 4

**Review:**

This paper utilizes ACOL algorithm for unsupervised learning. ACOL can be considered a type of semi-supervised learning where the learner has access only to parent-class information (for example in digit recognition whether a digit is bigger than 5 or not) and not the sub-class information (number between 0-9). Given that in many applications such parent-class supervised information is not available, the authors of this paper propose domain specific pseudo parent-class labels (for example transformed images of digits) to adapt ACOL for unsupervised learning. The authors also modified affinity and balance term utilized in GAR (as part of ACOL algorithm) to improve it. The authors use multiple data sets to study different aspects of the proposed approach.

I updated my scores based on the reviewers responses. It turned out that ACOL and GAR are also originally proposed by the same authors and was only published in arxiv! Because of the double-blind review nature of ICLR, I didn't know these ideas came from the same authors and is being published for the first time in a peer-reviewed venue (ICLR). So my main problem with this paper, lack of novelty, is addressed and my score has changed. Thanks to the reviewer for clarifying this.

---

> ### Author Response · Authors · 2017-12-23
> **Author feedback**
>
> We’d like to thank the reviewer for their time spent reviewing the paper and their valuable feedback and comments.  We wanted to address the two specific comments on the approach being incremental and the number of datasets.
>
> Both methods described in the article, ACOL and GAR are completely novel and this paper, if chosen for publication, will be the very first time they appear in peer-reviewed literature (a major reason why we chose ICLR).  Their adaptation to unsupervised settings – is also completely novel by extension with domain specific transformation a key factor in clustering performance.
>
> In this paper, we have actually used three (not one as suggested by the reviewer) image datasets MNIST, USPS and SVHN for comparison chosen primarily for uniformity and clarity between this article and many other recent ones published in conferences just like this one, ICLR, ICML, NIPS etc. MNIST and USPS datasets (hand-written digits) might be seen as simple datasets since the existing methods in the literature of clustering have already achieved very good performances on these two datasets. However, they are still used very commonly for benchmarks, especially on significantly different approaches such as the one proposed here.  More importantly, unlike semi-supervised and supervised settings, SVHN (a more realistic dataset with colored street view house numbers) still constitutes a very challenging task for unsupervised settings. This difficulty might be hard to observe in the first version of our paper as at the time of submission for ICLR 2018 we weren’t able to find any other work studying this dataset.
>
> Thanks to one of the commenters on the paper, we looked at IMSAT[1] as the previous state-of-art approach for clustering on SVHN, which was very recently published in ICML 2017 (a month before the submission deadline for ICLR – a reason why it wasn’t included in the first version). They have also presented the performances of other approaches, such as DEC[2], on the challenging SVHN dataset. Please see the clustering performances of two approaches reported by IMSAT[1] compared to the proposed approach below:
>
> DEC: 11.9%(±0.40)
> IMSAT: 57.3%(±3.90)
> Our Approach: 76.8%(±1.30)
>
> We will include this new article and believe this comparison would further reinforce the state-of-the-art capability and accurateness of our approach on a multitude of datasets.
>
> ----------------------------------------------------------------------------------------------------------------------------------------------------
>
> [1]: Learning Discrete Representations via Information Maximizing Self-Augmented Training,
> author    = {Weihua Hu, Takeru Miyato, Seiya Tokui, Eiichi Matsumoto, Masashi Sugiyama},
> booktitle = {Proceedings of the 34th International Conference on Machine Learning,
>                {ICML} 2017, Sydney, NSW, Australia, 6-11 August 2017}
>
> [2]: Unsupervised Deep Embedding for Clustering Analysis},
>   author    = {Junyuan Xie, Ross B. Girshick, Ali Farhadi},
>   booktitle = {Proceedings of the 33nd International Conference on Machine Learning,
>                {ICML} 2016, New York City, NY, USA, June 19-24, 2016}

---

### Official Review · AnonReviewer2 · 2017-12-02
**This paper presents a method for clustering based on latent representations learned from the classification of transformed data after pseudo-labellisation corresponding to applied transformation.**

**Rating:** 6
**Confidence:** 5

**Review:**

This paper presents a method for clustering based on latent representations learned from the classification of transformed data after pseudo-labellisation corresponding to applied transformation. Pipeline: -Data are augmented with domain-specific transformations. For instance, in the case of MNIST, rotations with different degrees are applied. All data are then labelled as "original" or "transformed by ...(specific transformation)". -Classification task is performed with a neural network on augmented dataset according to the pseudo-labels. -In parallel of the classification, the neural network also learns the latent representation in an unsupervised fashion. -k-means clustering is performed on the representation space observed in the hidden layer preceding the augmented softmax layer.

Detailed Comments:
(*) Pros
-The method outperforms the state-of-art regarding unsupervised methods for handwritten digits clustering on MNIST.
-Use of ACOL and GAR is interesting, also the idea to make "labeled" data from unlabelled ones by using data augmentation.

(*) Cons
-minor: in the title, I find the expression "unsupervised clustering" uselessly redundant since clustering is by definition unsupervised.
-Choice of datasets: we already obtained very good accuracy for the classification or clustering of handwritten digits. This is not a very challenging task.
And just because something works on MNIST, does not mean it works in general.
What are the performances on more challenging datasets like colored images (CIFAR-10, labelMe, ImageNet, etc.)?
-This is not clear what is novel here since ACOL and GAR already exist. The novelty seems to be in the adaptation to GAR from the semi-supervised to the unsupervised setting with labels indicating if data have been transformed or not.


My main problem  was about the lack of novelty. The authors clarified this point, and it turned out that ACOL and GAR have never published elsewhere except in ArXiv.  The other issue concerned the validation of the approach on databases other than MNIST. The author also addressed this point, and I changed my scores accordingly.

---

> ### Author Response · Authors · 2017-12-23
> **Author feedback**
>
> We’d like to thank the reviewer for their time spending reviewing the paper and their valuable feedback and comments.  We wanted to address the comments where we thought the reviewer could use some clarification especially in regards to the novelty of the approach.
>
> In response to choice of wording in title:
>
> We agree with the reviewer that the word unsupervised can be removed from the title while preserving the same meaning.
>
> In response to the choice of datasets:
>
> We believe the state-of-the-art performance of the proposed approach on the SVHN dataset may not have been properly emphasized in the article.
>
> In this paper, we used three image datasets (not just MNIST as suggested by the reviewer) MNIST, USPS and SVHN for comparison chosen primarily for uniformity and clarity between this article and many other recent ones published in conferences just like this one, ICLR, ICML, NIPS etc. MNIST and USPS datasets (hand-written digits) might be seen as simple datasets since the existing methods in the literature of clustering have already achieved very good performances on these two datasets. However, they are still used very commonly for benchmarks, especially on significantly different approaches such as the one proposed here.  More importantly, unlike semi-supervised and supervised settings, SVHN (a more realistic dataset with colored street view house numbers) still constitutes a very challenging task for unsupervised settings. This difficulty might be hard to observe in the first version of our paper as at the time of submission for ICLR 2018 we weren’t able to find any other work studying this dataset.
>
> Thanks to one of the commenters on the paper, we looked at IMSAT[1] as the previous state-of-art approach for clustering on SVHN, which was very recently published in ICML 2017 (a month before the submission deadline for ICLR – a reason why it wasn’t included in the first version). They have also presented the performances of other approaches, such as DEC[2], on the challenging SVHN dataset. Please see the clustering performances of two approaches reported by IMSAT[1] compared to the proposed approach below:
>
> DEC: 11.9%(±0.40)
> IMSAT: 57.3%(±3.90)
> Our Approach: 76.8%(±1.30)
>
> We will include this new article and believe this comparison would further reinforce the state-of-the-art capability and accurateness of our approach.
>
> Finally, to the best of our knowledge, clustering on the datasets such as CIFAR-10, labelMe, ImageNet based on their raw pixel values is not a very common practice in the literature of clustering as raw pixels are not suited for this goal with color information being dominant [1]. Existing approaches perform the clustering on the extracted features from these datasets. However, this approach doesn’t fit the proposed clustering technique in this paper, because transformations generating the pseudo classes are domain-specific and so they are directly applied on the input space. Therefore, generalizing the proposed clustering technique to these datasets requires an orthogonal challenge which we already identified as future work to study how to apply these domain-specific transformations that will present rich information for the clustering task at hand.
>
> In response to the comments about novelty:
>
> We don’t think this comment – quote “This is not clear what is novel here since ACOL and GAR already exist. The novelty seems to be in the adaptation to GAR from the semi-supervised to the unsupervised setting with labels indicating if data have been transformed or not”  reflects the reality as both methods ACOL and GAR are completely novel and this paper, if chosen for publication, will be the very first time they appear in peer-reviewed literature (a major reason why we chose ICLR).  Their adaptation to unsupervised settings – is also completely novel by extension with domain specific transformation a key factor in clustering performance.
>
> ----------------------------------------------------------------------------------------------------------------------------------------------------
>
> [1]: Learning Discrete Representations via Information Maximizing Self-Augmented Training,
> author    = {Weihua Hu, Takeru Miyato, Seiya Tokui, Eiichi Matsumoto, Masashi Sugiyama},
> booktitle = {Proceedings of the 34th International Conference on Machine Learning,
>                {ICML} 2017, Sydney, NSW, Australia, 6-11 August 2017}
>
> [2]: Unsupervised Deep Embedding for Clustering Analysis},
>   author    = {Junyuan Xie, Ross B. Girshick, Ali Farhadi},
>   booktitle = {Proceedings of the 33nd International Conference on Machine Learning,
>                {ICML} 2016, New York City, NY, USA, June 19-24, 2016}

---

### Public Comment · (anonymous) · 2017-11-11
**Request for citation**

I believe that you should also cite “Learning Discrete Representations via Information Maximizing Self-Augmented Training” (ICML 2017) http://proceedings.mlr.press/v70/hu17b.html.
This paper is closely related to your work and is also about unsupervised clustering using deep neural networks.
As far as I know, the proposed method, IMSAT, is the current state-of-the-art method in deep clustering (November 2017). Could you compare your results against their result?

---

> ### Author Response · Authors · 2017-11-11
> **Citing IMSAT and comparison**
>
> First of all, we would like to thank you for your comment.
>
> We'll cite this work in the upcoming revision. But I think revisions are currently not allowed as the review process is ongoing.
>
> So as a quick answer to your comment,  I can say that it's hard to observe any statistically significant differences between the performances of these two models on the MNIST dataset. That is, 98.4 (0.4) for IMSAT vs. 98.32%(±0.08) for our approach. But the SVHN dataset provides a more solid basis of comparison. While IMSAT achieves 57.3 (3.9) clustering accuracy on SVHN, our approach outperforms IMSAT by achieving 76.80%(±1.30). I think it's also worth noting the mechanical differences between these two approaches. For example, while IMSAT uses 960-dimensional GIST features for SVHN, our approach employs raw pixels (32x32x3). Besides, IMSAT uses VAT based (or RPT based) regularization while we adopt graph-based regularization.
>
> I think citing this work will further be helpful for the evaluation of our approach, as it also provides the performances of other approaches (like DEC) on the SVHN dataset.
>
> Thank you very much for your feedback.

---

### Public Comment · (anonymous) · 2017-12-05
**Why don't compare with the baselines on other datasets, such as Reuters?**

All the datasets in the paper are quite simple. The paper should compare with other baselines on Reuters, since it is much bigger.

---

> ### Author Response · Authors · 2017-12-19
> **Comparison on the Reuters dataset**
>
> First of all, we would like to thank you for your comment.
>
> Very briefly, we haven't used the Reuters dataset (or any other text categorization dataset) because generalizing our approach to sequential domain requires a future work (and possibly a new article) to study the domain-specific transformations that will present a useful knowledge about the clustering task at hand on this domain.
>
> In this paper, we used three image datasets MNIST, USPS and SVHN for the comparison. MNIST and USPS datasets (hand-written digits) might be seen as simple datasets since the existing methods in the literature of clustering have already achieved very good performances on these two datasets. However, unlike for semi-supervised and supervised settings, SVHN (street view house numbers - more realistic) is still a difficult and unsolved problem for the clustering task. This difficulty might be hard to observe in the first version of our paper as we couldn't find a method reporting clustering performance on SVHN to compare our approach at the time of submission for ICLR 2018.
>
> IMSAT[1] is the previous state-of-art approach for clustering on  SVHN, which was very recently published on ICML 2017 (the reason we missed this method in the first version). They also presented the performances of other approaches, such as DEC[2], on the SVHN dataset. Please see the clustering performances of two approaches reported by IMSAT[1] and also our approach below.
>
> DEC:                    11.9%(±0.40)
> IMSAT:                57.3%(±3.90)
> Our Approach:  76.8%(±1.30)
>
> We think this comparison would help the reader better observe the capability and accurateness of our approach.
>
>
> Besides, please note the performances of DEC[2] and IMSAT[1] on Reuters.
> DEC:                    67.3%(±0.20)
> IMSAT:                71.0%(±4.90)
>
> By comparing the performances of these two models on the SVHN and on the Reuters datasets, we believe it is not totally wrong to say that SVHN presents a harder problem for clustering. We just want to note that not using the Reuters dataset for the comparison is not due to its complexity or largeness but due to the requirement for a future work for generalizing our approach to other domains such as text categorization. However, there is, of course, no guarantee that our approach will perform well on other domains.
>
> Thank you very much for your feedback.
>
> [1]: Learning Discrete Representations via Information Maximizing Self-Augmented Training,
> author    = {Weihua Hu, Takeru Miyato, Seiya Tokui, Eiichi Matsumoto, Masashi Sugiyama},
> booktitle = {Proceedings of the 34th International Conference on Machine Learning,
>                {ICML} 2017, Sydney, NSW, Australia, 6-11 August 2017}
>
> [2]: Unsupervised Deep Embedding for Clustering Analysis},
>   author    = {Junyuan Xie, Ross B. Girshick, Ali Farhadi},
>   booktitle = {Proceedings of the 33nd International Conference on Machine Learning,
>                {ICML} 2016, New York City, NY, USA, June 19-24, 2016}

---

> > ### Public Comment · (anonymous) · 2017-12-20
> > **There is no need to generalize your method to sequential domain on Reuters**
> >
> > Thanks for the reply. You mentioned that "we haven't used the Reuters dataset (or any other text categorization dataset) because generalizing our approach to sequential domain requires a future work (and possibly a new article) ". However, as far as I know, DEC also report the results on Reuters datasets and you can use the Bag of Words to represent the documents. Hence, you do not need to use sequential methods and you can compare with the baselines directly.

---

> > > ### Author Response · Authors · 2018-01-04
> > > **Using Bag of Words representation**
> > >
> > > Even if we use the Bag of Words to represent the documents, the proposed approach still needs one or more useful transformations (except the non-transformation T1 in equation 8) to create other pseudo-classes as expressed in equation 8 in the manuscript. If you have any suggestions about the transformations that can be applied to the Bag of Words representation, we'd like to apply them during our studies for the future work on expanding this approach to the sequential domain.

---

### Author Response · Authors · 2017-12-23
**Summary of author feedbacks for reviewers' comments**

We wanted to thank the reviewers for their time spent on our article and both their encouraging remarks and valuable feedback.  Two of the most critical comments in regards to novelty and choice of datasets were addressed separately in response to individual reviewers.  In summary, the reported state-of-the-art results and the datasets chosen for this study (three, not one as one reviewer suggested) mirrors those published in reputable venues such as ICLR, ICML, NIPS, etc. for direct comparison.  In addition, both the framework (GAR + ACOL) and its extension to unsupervised learning are completely novel and have not appeared in any peer-reviewed publication until now. More detailed explanations can be found under responses to each reviewer.

---

### Author Response · Authors · 2018-01-04
**Revision 2 and the summary of the changes**

Following minor changes have been made in the Revision 2 of this article. Please also note that the same manuscript has been mistakenly submitted three times while uploading the Revision 2.  So you can consider the latest submission in the Revision History as the Revision 2 during the pdf diff.

1. We added the results of IMSAT (Hu et al., 2017) in Table 3 (Quantitative unsupervised clustering performance in terms of ACC score) for the comparison and revising the corresponding comments about this table. That is,

"Our approach statistically significantly outperforms all the contemporary methods that reported unsupervised clustering performance on MNIST except IMSAT (Hu et al., 2017) displaying very competitive performance with our approach, i.e. 98.32%(+/-0.08) vs. 98.40%(+/-0.40). However, results obtained on the SVHN dataset, i.e. 76.80%(+/-1.30) vs. 57.30%(+/-3.90), show that our approach statistically significantly outperforms IMSAT on this realistic dataset and defines the current state-of-the-art for unsupervised clustering on SVHN."

2. We removed Table 4 (Quantitative unsupervised clustering performance in terms of NMI score) because, out of 9 approaches used for the performance comparison, only one of them reported NMI score. So, this table has been removed for the sake of simplicity as it introduces no further information than Table 3.

------------------------------------------------------------------------------------------------------------

[Hu et al., 2017] : Learning Discrete Representations via Information Maximizing Self-Augmented Training,
author    = {Weihua Hu, Takeru Miyato, Seiya Tokui, Eiichi Matsumoto, Masashi Sugiyama},
booktitle = {Proceedings of the 34th International Conference on Machine Learning,
               {ICML} 2017, Sydney, NSW, Australia, 6-11 August 2017}

---

### Decision · Program_Chairs · 2018-01-29
**ICLR 2018 Conference Acceptance Decision**

**Decision:**

Accept (Poster)

**Comment:**

The reviewers concerns regarding novelty and the experimental evaluation have been resolved accordingly and all recommend acceptance. I would recommend removing the term "unsupervised" in clustering, as it is redundant. Clustering is, by default, assumed to be unsupervised.

There is some interest in extending this to non-vision domains, however this is beyond the scope of the current work.